# Does the Correctness of Factual Knowledge Matter for Factual Knowledge-Enhanced Pre-trained Language Models?

**Boxi Cao**[1,3*], **Qiaoyu Tang**[1,3*], **Hongyu Lin**[1†], **Xianpei Han**[1,2†], **Le Sun**[1,2]

[1]Chinese Information Processing Laboratory    [2]State Key Laboratory of Computer Science
Institute of Software, Chinese Academy of Sciences, Beijing, China
[3]University of Chinese Academy of Sciences, Beijing, China
`{boxi2020,tangqiaoyu2020,hongyu,xianpei,sunle}@iscas.ac.cn`

## Abstract

In recent years, the injection of factual knowledge has been observed to have a significant *positive correlation* to the downstream task performance of pre-trained language models. However, existing work neither demonstrates that pre-trained models successfully learn the injected factual knowledge nor proves that there is a *causal relation* between injected factual knowledge and downstream performance improvements. In this paper, we introduce a counterfactual-based analysis framework to explore the causal effects of factual knowledge injection on the performance of language models within pretrain-finetune paradigm. Instead of directly probing the language model or exhaustively enumerating potential confounding factors, we analyze this issue by perturbing the factual knowledge sources at different scales and comparing the performance of pre-trained language models before and after the perturbation. Surprisingly, throughout our experiments, we find that although the knowledge seems to be successfully injected, the correctness of injected knowledge only has a very limited effect on the models' downstream performance. This finding strongly challenges previous assumptions that the injected factual knowledge is the key for language models to achieve performance improvements on downstream tasks in pretrain-finetune paradigm.

## 1 Introduction

In recent years, pre-trained language models (PLMs) have emerged as the dominant approach in natural language processing. Through self-supervised learning on large-scale text corpus, PLMs can acquire different kinds of knowledge automatically without additional manual guidance, which demonstrates significant generalizability and

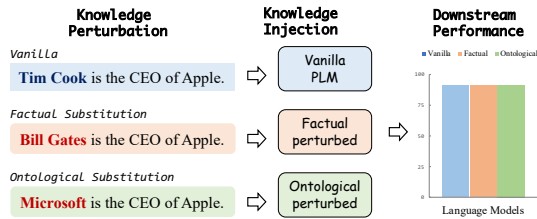

Figure 1: This paper explores the impact of factual knowledge by comparing the downstream task performance before and after knowledge perturbation.

transferability improvements across tasks compared with previous architectures (Devlin et al., 2019; Radford et al., 2019; Liu et al., 2019; Raffel et al., 2020; Scao et al., 2022; Touvron et al., 2023).

Some previous investigations contribute the superiors of pre-trained language models to their entailed various kinds of knowledge learned from the pre-training stage (Petroni et al., 2019; Lewis et al., 2020b; Yin et al., 2022; Cao et al., 2023). Among them, factual knowledge, which reveals the relationships between real-world entities (e.g., Tim Cook is the CEO of Apple) and plays a critical role in human cognition (Unger, 1968), is regarded as a critical factor for a pre-trained language model to approach a trusted intelligent agent (Lewis et al., 2020b; Yin et al., 2022). Consequently, how to improve the acquisition, modeling, and application of factual knowledge of pre-training language models has become a hot research topic. To this end, many studies have been devoted to further injecting factual knowledge to enhance the acquisition and modeling of factual knowledge in pre-trained language models (Zhang et al., 2019; Liu et al., 2020; Sun et al., 2020; Wang et al., 2021a,b), and have already reported successful performance improvements on specific downstream tasks.

On the contrary, recent studies have found that pre-trained language models struggle with retaining factual knowledge, and the retained factual knowledge can exhibit inconsistencies with the original knowledge sources (Poerner et al., 2020; Elazar

---

*Equal contributions.
†Corresponding authors.

et al., 2021; Cao et al., 2021). Furthermore, the indirect evaluations through downstream tasks only reflect that injecting factual knowledge is *correlated* to performance improvement, but can not establish the *causality* between them due to the existence of many additional potential confounding factors (training data domain, model parameter size, etc.). Consequently, to identify the impact of factual knowledge injection on the downstream performance of pre-trained language models, the following two critical questions should be answered:

- Through the existing knowledge injection methods, is the factual knowledge really injected into pre-trained language models?

- If so, is it indeed the injected knowledge, rather than other confounding factors, that is responsible for the observed performance improvements in downstream tasks?

Unfortunately, it is infeasible to directly answer the above-two questions due to the lack of highly effective language model knowledge probing and confounding factor identification measurements. To this end, as shown in Figure 1, this paper introduces a counterfactual-based analysis framework (Veitch et al., 2021; Guidotti, 2022) to explore the causal effects of injecting factual knowledge in a "what-if" manner. Moreover, Figure 2 illustrates the applied framework. Instead of directly probing the language model or exhaustively enumerating potential confounding factors, we analyze the above-two questions by perturbing the factual knowledge sources at different scales, then comparing the downstream performance of pre-trained language models before and after the perturbation. The key motivation of our work is that: 1) If the knowledge injection approaches are ineffective, the performance of the model injected with the correct knowledge and perturbed knowledge should not perform very differently in the knowledge probing evaluation; 2) If the correctness of injected factual knowledge is indeed essential for downstream tasks, then injecting perturbed, wrong knowledge should cause significant performance decline. Specifically, in order to observe the aspect from which factual knowledge affects PLMs, we conduct two kinds of perturbation on factual knowledge, including *factual substitution*, which replaces an entity in factual knowledge with another entity with the same type (e.g., substitute "*Tim Cook* is the CEO of Apple" with "*Bill Gates*

is the CEO of Apple"), as well as *ontological substitution* that thoroughly perturb entities with their counterparts of another type (e.g., substitute "*Tim Cook* is the CEO of Apple" with "*Microsoft* is the CEO of Apple"). In addition, to analyze the impact of perturbation, we investigate three knowledge injection approaches that acquire knowledge from two types of factual knowledge sources, including plain texts containing factual knowledge and structured factual knowledge bases.

Throughout empirical evaluations on a wide range of representative downstream NLP tasks, our findings surprisingly deviate from previous hypotheses. Although the knowledge injection approaches seem to inject factual knowledge into PLMs successfully, we find that factual knowledge perturbation has only a very limited effect on PLMs' downstream task performance, i.e., the correctness of factual knowledge shows a very limited impact on all evaluated downstream tasks. Furthermore, although the influence of ontological perturbation is slightly stronger than factual perturbation, it also does not cause statistically significant performance divergence in most downstream tasks. Specifically, our experiments show that in most downstream tasks, the performance fluctuation caused by the above-two perturbation is not greater than the fluctuation caused by random seeds, and the results of t-test further demonstrate that there are no statistically significant differences between the performance before and after perturbation. Through this counterfactual-based analysis, our findings demonstrate that injected factual knowledge is not the core reason for the performance improvements of previous knowledge injection approaches on pre-trained language models in the pretrain-finetune paradigm.

The following part of this paper is organized as follows. In Section 2, we briefly review the related work about factual knowledge probing and injection of PLMs. Section 3 presents our proposed counterfactual-based analysis framework. The experimental results and the process leading to our conclusions are presented in Section 4. In Section 5, we provide a brief discussion and conclude our findings[1].

---

[1] We openly released our source code at `https://github.com/tangqiaoyu/KnowledgeDisturb`

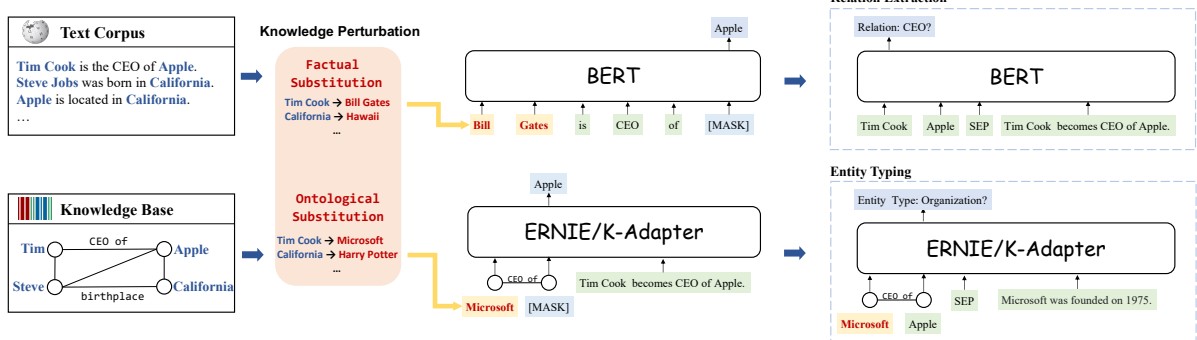

Figure 2: The illustration of the counterfactual-based knowledge analysis framework. Given factual knowledge from text corpus or knowledge bases, we first conduct two perturbations to obtain incorrect knowledge instances. Then we pre-train LMs on the perturbed datasets. Finally, we compare the downstream performance of PLMs before and after perturbation to explore the causal effects of factual knowledge injection on the performance of PLMs.

## 2 Related Work

Factual knowledge reveals the relationships between real-world entities and plays a crucial role in human cognitive (Unger, 1968). Therefore, lots of studies devoted to probing the factual knowledge entailed in PLMs on the one hand (Petroni et al., 2019; Kassner et al., 2021; Sung et al., 2021), and propose to enhance the factual knowledge in large-scale LMs to benefit their performance on downstream tasks on the other (Zhang et al., 2019; Liu et al., 2020; Wang et al., 2021b).

**Factual Knowledge Probing** aims to evaluate how well current PLMs are entailing factual knowledge in parameters. Currently, the most popular strategy is prompt-based probing (Petroni et al., 2019; Ettinger, 2020; Srivastava et al., 2022), e.g., query BERT with "Tim Cook is the CEO of [MASK]" to determine whether it contains corresponding knowledge. And recent studies have found that prompt-based probing could be inaccurate (Poerner et al., 2020; Zhong et al., 2021), inconsistent (Elazar et al., 2021; Kassner and Schütze, 2020; Cao et al., 2022; Jang et al., 2022), and unreliable (Li et al., 2022; Cao et al., 2021).

**Factual Knowledge Injection** aims to inject factual knowledge into PLMs. Currently, there are two main sources for injecting factual knowledge into language models, including plain text and structured knowledge base. For plain text, LMs typically acquire factual knowledge through self-supervised learning on large-scale text corpus without any knowledge guided supervision. The popular objectives include casual language modeling (Radford et al., 2019; Brown et al., 2020; Scao et al., 2022; Touvron et al., 2023), masked language

modeling (Devlin et al., 2019; Liu et al., 2019), denoising auto-encoder (Lewis et al., 2020a), etc. And such pre-trained LMs have been shown to potentially entail a large scale of factual knowledge in the parameters (Bouraoui et al., 2020; Petroni et al., 2019). In addition, many studies intend to explicitly infuse the factual knowledge from structured knowledge base into LMs (Yang et al., 2021). Popular strategies include: 1) Embedding combined methods (Zhang et al., 2019; Peters et al., 2019; He et al., 2020b), which encode structured knowledge via knowledge embedding algorithms and then enhance PLMs' text representation with knowledge graph embedding; 2) Knowledge supervised methods (Wang et al., 2021a,b; Yamada et al., 2020), which utilize elements from structured knowledge as supervision signals and leverage base PLMs to learn their semantics.

## 3 Counterfactual-based Knowledge Analysis Framework

As mentioned above, this paper intends to investigate whether current factual knowledge injection approaches can inject factual knowledge into pre-trained language models and whether the injected factual knowledge can benefit downstream NLP tasks in pretrain-finetune paradigm. However, it is currently not feasible to directly explore these questions due to the lack of effective knowledge probing and confounding factor discovery approaches for pre-trained language models.

To this end, we propose to leverage a counterfactual-based knowledge analysis framework to answer these two questions in a "what-if" manner. Specifically, we transform the problem of investigating the characteristics of a model with

| Source | Wikipedia | Wikidata | |
|---|---|---|---|
| Model | BERT | ERNIE | K-Adapter |
| Total | 14,545,579 | 6,105,524 | 5,565,478 |
| Perturbed | 13,538,337 | 5,541,297 | 4,692,683 |
| Pert. Rate | 93.08% | 90.76% | 84.32% |

Table 1: The perturbation details of two kinds of knowledge sources. For text corpus, we perturb 93.1% paragraphs from Wikipedia. As for structured knowledge base, we perturb 90.76% training instances for ERNIE and 84.32% for K-Adapter from Wikidata.

injected knowledge into comparing the behaviors between models injected with correct and incorrect knowledge, respectively. Consequently, if the knowledge injection approaches work, models injected with incorrect knowledge should exhibit significantly inferior performance on the knowledge probing evaluation than models injected with correct knowledge. Furthermore, if injecting factual knowledge is indeed helpful to downstream tasks, models injected with correct factual knowledge should perform significantly better than models injected with incorrect factual knowledge on downstream tasks.

The overall counterfactual-based analysis framework is illustrated in Figure 2. Specifically, given factual knowledge from text corpus or knowledge bases, we first conduct perturbation on the instances in them to obtain incorrect knowledge instances. Then we pre-train language models with several representative factual knowledge injection approaches on both vanilla and perturbed knowledge sources. Finally, we compare the performance of knowledge-injected models to reach conclusions about the above-two questions. In the following sections, we will first demonstrate how we conduct knowledge perturbation on different knowledge sources and then briefly introduce the investigated representative factual knowledge injection approaches in our experiments.

### 3.1 Factual Knowledge Perturbation

Knowledge perturbation aims to generate counterfactual factual knowledge instances for analysis. In this paper, we mainly employ two kinds of perturbation strategies, including *factual substitution* and *ontological substitution*. Factual substitution studies the influence of the *factual correctness* of factual knowledge in models by replacing entities with other entities of the same type. For example, factual substitution perturbs the factual knowledge "*Tim Cook* is the CEO of Apple" with an incorrect

statement "*Bill Gates* is the CEO of Apple". On the other hand, ontological substitution thoroughly perturbs entities with counterparts of different types, which is used to evaluate the importance of factual knowledge ontology on downstream tasks. For example, ontological substitution replaces "*Tim Cook* is the CEO of Apple" with "*Microsoft* is the CEO of Apple".

This paper mainly focuses on two kinds of the most widely-used factual knowledge sources, including learning from plain text corpus and structural factual knowledge bases. For learning from plain text corpus, we use paragraphs in Wikipeida[2] that contain an anchor linking to Wikidata (Vrandečić and Krötzsch, 2014) as knowledge sources. For learning from knowledge bases, we direct use entities and their relations in Wikidata as knowledge sources. The type of an entity is determined by its corresponding "instance of" and "subclass of" properties in Wikidata. In this paper, we mainly focus on perturbing factual knowledge about three kinds of representative entity types, including Person, Location, and Organization. Table 1 demonstrates the perturbation details for both kinds of knowledge sources, revealing that the perturbation would affect most of their training instances. For learning from text, we utilize 14,545,579 paragraphs from Wikipedia and perturb 13,538,337 of them, resulting in a 93.1% perturbation rate. For knowledge learning from structured data such as ERNIE, we utilize a total of 6,105,524 pre-training instances, of which we perturb 5,531,297, leading to a perturbation rate of 90.76%.

### 3.2 Factual Knowledge Injection Approaches

In recent years, the injection of factual knowledge into large-scale language models has emerged as a prominent research area. Various methods have been developed for injecting knowledge, depending on the specific sources of knowledge. In this paper, we explore three representative approaches for injecting knowledge, corresponding to the following two types of knowledge sources:

**Learning From Plain Text.** In this study, we select BERT (Devlin et al., 2019) as our experiment architecture for learning from text, as it is one of the most representative pre-trained language models. To investigate the impact of factual knowledge on BERT, we conduct pre-training of the BERT-base model *from scratch* with masked language

[2]https://www.wikipedia.org/

modeling as the objective, separately on both the vanilla and perturbed versions of Wikipedia text corpus respectively. The model is pre-trained using a batch size of 1024 sequences for 500,000 steps. For optimization, we use Adam optimizer with a learning rate of $1e-4$, $\beta_1 = 0.9$, $\beta_2 = 0.999$, learning rate warmup over the first 10,000 steps. The training process was conducted on 2 Nvidia A100 GPUs with 80G RAM for about 10 days.

**Learning From Structured Knowledge.** We select one representative approach for each direction of learning factual knowledge from structured knowledge mentioned in Section 2, including:

- ERNIE (Zhang et al., 2019) is a typical model that injects knowledge into PLMs through the embedding combined method. It first identifies the named entity mentions in the text and aligns them to Wikidata. Then ERNIE aggregates the entity representation with its corresponding token embedding, where entity representation is trained on KG via knowledge embedding algorithms like TransE (Bordes et al., 2013). We perturb the acquired knowledge for ERNIE by substituting the entity representation in the input.

- K-Adapter (Wang et al., 2021a) is a representative method that utilizes elements in the explicit knowledge base as the supervision signal. It designs an adapter to inject factual knowledge via relation classification task with keeping the original parameters of PLM fixed. We perturb the acquired knowledge for K-Adapter by directly substituting the entities in the explicit knowledge base.

To ensure a fair comparison, we strictly adhere to the pre-training process outlined in the original papers.

### 3.3 Downstream Evaluation

To make a comprehensive and thorough evaluation, we conduct experiments on a wide range of downstream tasks, most of which have been previously shown to achieve performance improvement through knowledge injection (Zhang et al., 2019; Wang et al., 2021a; He et al., 2020a; Yamada et al., 2020; Qin et al., 2021; Sun et al., 2021). Inspired by Yu et al. (2023), we divide these tasks into four categories based on the stratification and connection to factual knowledge: knowledge probing tasks, knowledge guided tasks, knowledge applying tasks, and language understanding tasks.

**Knowledge Probing Tasks** are primarily used to investigate the knowledge entailed in PLMs. We use LAMA (Petroni et al., 2019), the most widely used factual knowledge probing benchmark, as our testbed to determine whether the factual knowledge is successfully injected into PLMs. LAMA evaluates the factual knowledge in PLMs by employing cloze-style questions, such as "Tim Cook is the CEO of [MASK]".

**Knowledge Guided Tasks** aim to evaluate the ability of PLMs to recognize the factual information within texts, such as entities and entity relations. Specifically, we evaluate BERT on two widely used named entity recognition (NER) datasets including CONLL2003 (Tjong Kim Sang, 2002) and OntoNotes 5.0 (Pradhan et al., 2013), as well as two representative relation extraction (RE) datasets including ACE2004 (Mitchell et al., 2005) and ACE2005 (Walker et al., 2006). For the evaluation of ERNIE and K-Adapter, to obtain more reliable experimental conclusions, we conduct experiments on *the same tasks as the original paper*, including entity typing (e.g., Open Entity (Choi et al., 2018) and FIGER (Ling et al., 2015)) and relation classification (e.g., FewRel (Han et al., 2018) and TACRED (Zhang et al., 2017)).

**Knowledge Applying Tasks** focus on evaluating the model's ability to apply factual knowledge to reasoning and problem-solving tasks (Petroni et al., 2021), e.g., open QA and fact checking. In this paper, we select the open QA datasets Natural Questions (Kwiatkowski et al., 2019), CosmosQA (Huang et al., 2019), and fact checking dataset FEVER (Thorne et al., 2018).

**Language Understanding Tasks** includes various tasks such as text classification, natural language inference, sentiment analysis, etc. We select GLUE (Wang et al., 2019) as evaluation benchmark, which is a collection of NLU tasks and widely used in various PLMs' evaluations, and previous knowledge injection studies have reported performance improvements on them (Sun et al., 2019; Liu et al., 2020; Sun et al., 2021).

In the downstream task fine-tuning process, we follow the same dataset split and hyper-parameter settings of the original papers, please refer to Appendix for details due to page limitation. In addition, to avoid the impact of randomness on investigation conclusions, all experiments were conducted

| | Perturbation | | | | Factual Query | Output |
|---|---|---|---|---|---|---|
| Factual | Cicero | ⟶ | Lorde | vanilla | **Cicero** was born in [MASK] . | Rome |
| | Rome | ⟶ | Disney | perturb | **Lorde** was born in [MASK] . | Disney |
| Ontological | Mahatma Gandhi | ⟶ | Luzon | vanilla | **Mahatma Gandhi** was born in [MASK] . | India |
| | India | ⟶ | Nevada | perturb | **Luzon** was born in [MASK] . | Nevada |

Table 2: Example of PLMs' outputs before and after perturbation. The left part shows the perturbation mapping and the right shows the input and output of the models; "factual" and "ontological" denotes the corresponding substitution strategy; "vanilla" and "perturb" denote the vanilla LM and the perturbed LM respectively.

| Model | Vanilla | Factual | Ontological |
|---|---|---|---|
| BERT | 28.18 | 11.62 | 10.34 |
| ERNIE | 29.21 | 25.31 | 25.67 |

Table 3: The factual knowledge probing results on LAMA for both vanilla and perturbed pre-trained language models.

under 5 random seed settings, and their means and standard deviations are reported in the later results.

## 4 Experiments and Findings

Based on the analysis framework presented in Section 3, we conduct extensive experiments and intriguingly find that the correctness of factual knowledge shows a very limited impact on almost all downstream tasks. In this section, we will elaborate on our experimental procedures, provide a detailed account of our findings, and illustrate how we arrive at our conclusions.

### 4.1 Does Knowledge Injection Works?

**Conclusion 1.** *Knowledge injection approaches successfully affect factual knowledge in pre-trained language models.*

To assess the effectiveness of knowledge injection approaches, we compare the performance of PLMs before and after perturbation on the well-known knowledge probing benchmark LAMA (Petroni et al., 2019). Table 2 demonstrates several illustrative examples of the predictions from both vanilla and perturbed models, showcasing the influence of the knowledge injection process on the model's output. For example, in the perturbed corpus, we substitute the factual knowledge <Cicero, birthplace, Rome> with <Lorde, birthplace, Disney>. The vanilla BERT predicts "Cicero was born in Rome", while the perturbed BERT predicts "Lorde was born in Disney", indicating that factual knowledge injected during pre-training indeed influences the model's prediction.

To further quantify such effects, Table 3 shows

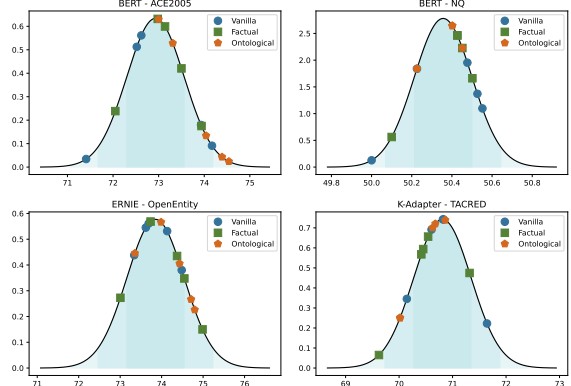

Figure 3: Distributions of LMs' performance on downstream tasks. We assume the variation in performance caused by random seed follows a normal distribution, and estimate the parameters based on the results of the vanilla language model on various random seeds, and the remaining points on the curve demonstrate the performance of LMs before and after perturbation.

the performance on LAMA benchmark for PLMs with and without different types of perturbation. It is evident that BERT's performance on LAMA significantly decreases from 28.18 to 11.62 after factual substitution and further drops to 10.34 after ontological substitution. ERNIE's performance drops from 29.21 to 25.31 and 25.67 after perturbation. The significant decline in performance demonstrates the effectiveness of injecting incorrect knowledge. In conclusion, through factual knowledge probing, we can demonstrate the effectiveness of knowledge injection on both learning factual knowledge from plain text and structural knowledge bases. This implies that current representative knowledge injection approaches successfully influence factual knowledge in language models.

### 4.2 Does Factual Substitutions Affect Downstream Performance?

**Conclusion 2.** *Regardless of the approaches of knowledge injection, factual substitution shows very limited influence on all downstream tasks, i.e.,*

| Model | Perturbation | CONLL2003 | OntoNotes 5.0 | ACE2004 | ACE2005 | NQ | FEVER | GLUE |
|---|---|---|---|---|---|---|---|---|
| BERT | No Substitution | $91.37 \pm 0.31$ | $88.92 \pm 0.10$ | $72.18 \pm 0.52$ | $72.93 \pm 1.01$ | $50.36 \pm 0.21$ | $88.58 \pm 0.24$ | $80.26 \pm 0.33$ |
| | Factual Substitution | $91.22 \pm 0.05$ | $88.87 \pm 0.11$ | $71.79 \pm 0.39$ | $73.12 \pm 0.63$ | $50.38 \pm 0.14$ | $88.40 \pm 0.18$ | $80.07 \pm 0.53$ |
| | P-value | 0.385 | 0.542 | 0.265 | 0.755 | 0.849 | 0.273 | 0.560 |

| Model | Perturbation | Open Entity | FIGER | FewRel | TACRED | GLUE |
|---|---|---|---|---|---|---|
| ERNIE | No Substitution | $73.85 \pm 0.41$ | $71.43 \pm 1.25$ | $88.41 \pm 0.33$ | $66.86 \pm 0.85$ | $81.74 \pm 0.59$ |
| | Factual Substitution | $74.13 \pm 0.69$ | $71.16 \pm 0.96$ | $86.88 \pm 0.34$ | $66.31 \pm 0.45$ | $81.57 \pm 0.34$ |
| | P-value | 0.509 | 0.740 | 0.000 | 0.291 | 0.638 |

| Model | Perturbation | Open Entity | FIGER | TACRED | CosmosQA | GLUE |
|---|---|---|---|---|---|---|
| K-Adapter | No Substitution | $76.05 \pm 0.36$ | $77.67 \pm 0.67$ | $70.81 \pm 0.49$ | $80.04 \pm 0.73$ | $87.31 \pm 0.75$ |
| | Factual Substitution | $76.05 \pm 0.43$ | $77.08 \pm 0.74$ | $70.47 \pm 0.54$ | $79.52 \pm 1.05$ | $87.90 \pm 0.56$ |
| | P-value | 0.995 | 0.276 | 0.376 | 0.437 | 0.240 |

Table 4: Model performance and t-test results on downstream tasks before and after factual substitution. The last row presents p-values from the t-test. Most p-values significantly exceed threshold 0.05, indicating insufficient statistical evidence to assert a noteworthy difference in model performance before and after perturbation.

*the correctness of injected factual knowledge is not the key factor for factual knowledge-enhanced language models to achieve better performance.*

To investigate the effect of knowledge perturbation, we calculate the mean and standard deviation of model performance with 5 different random seeds for each task and compare the performance fluctuation caused by perturbation and random seeds. Moreover, we leverage the t-test to further verify whether a statistically significant difference exists between the model performance before and after perturbation.

Table 4 shows the models' performance on downstream tasks before and after factual substitution. For learning from plain texts, we observe that factual substitution has a limited impact on all evaluated dataset: 1) On language understanding tasks using GLUE benchmark, the average performance fluctuation caused by factual substitution is $0.19\%$, which is lower than the performance fluctuation $0.33\%$ caused by random seeds. And the performance on each task is demonstrated in the appendix due to page limitations; 2) On knowledge applying tasks such as open domain QA and fact checking, we obtain similar findings to GLUE benchmark, indicating that the knowledge acquired from the pre-training phase has a limited impact even when the downstream tasks require specific knowledge for answer inference or task solving; 3) Even for the factual knowledge guided tasks such as NER and RE, we surprisingly find that the random seed still causes larger performance fluctuation than factual substitution. Moreover, on relation extraction tasks such as ACE 2005, the average performance of the perturbed models across different random seeds is even higher than the vanilla model without perturbation.

For learning from structured knowledge, we conduct experiments on both embedding combined (ERNIE) and knowledge supervised (K-Adapter) methods. In order to ensure the reliability of our conclusions, we select the same tasks as the original papers, where knowledge injection was shown to provide benefits. Overall, we reach similar findings with learning from plain texts, where the random seeds cause larger performance fluctuation than factual substitution on most benchmarks. The only exception comes from FewRel, where the factual substitution leads to relatively significant performance degeneration. However, it is worth noting that FewRel and ERNIE share the same knowledge source, which could lead to significant information leakage and make the model more dependent on the correctness of knowledge in the explicit knowledge base. We also conduct detailed experiments to prove and quantify the information leakage, which is beyond the scope of this paper, and therefore we present the results in the appendix.

To further quantify the performance divergence, we employ a t-test (Boneau, 1960) to examine the significance of the performance differences between the models before and after factual substitution. The null hypothesis posited no alteration in the mean performance, while the alternative hypothesis argued for a discernible variation in performance levels. Following standard conventions, we set the threshold for statistical significance at 0.05 That is to say, a p-value greater than 0.05 indicates that there is no sufficient statistical evidence to reject the null hypothesis. The p-values of the models on each downstream datasets are presented in the last row of Table 4. In all of these datasets (except FewRel as we mentioned above), the p-values were notably larger than our pre-specified level of statis-

| Model | Perturbation | CONLL2003 | OntoNotes 5.0 | ACE2004 | ACE2005 | NQ | FEVER | GLUE |
|-------|-------------|-----------|---------------|---------|---------|-----|-------|------|
| BERT | No Substitution | $91.37 \pm 0.31$ | $88.92 \pm 0.10$ | $72.18 \pm 0.52$ | $72.93 \pm 1.01$ | $50.36 \pm 0.21$ | $88.58 \pm 0.24$ | $80.26 \pm 0.33$ |
| | Ontological Substitution | $91.18 \pm 0.09$ | $88.72 \pm 0.15$ | $72.56 \pm 0.33$ | $73.86 \pm 0.60$ | $50.34 \pm 0.10$ | $88.23 \pm 0.28$ | $80.18 \pm 0.36$ |
| | P-value | 0.290 | 0.061 | 0.250 | 0.154 | 0.902 | 0.089 | 0.758 |

| Model | Perturbation | Open Entity | FIGER | FewRel | TACRED | GLUE |
|-------|-------------|-------------|-------|--------|--------|------|
| ERNIE | No Substitution | $73.85 \pm 0.41$ | $71.43 \pm 1.25$ | $88.41 \pm 0.33$ | $66.86 \pm 0.85$ | $81.74 \pm 0.59$ |
| | Ontological Substitution | $74.26 \pm 0.53$ | $70.00 \pm 1.60$ | $85.15 \pm 0.46$ | $66.57 \pm 0.35$ | $81.67 \pm 0.16$ |
| | P-value | 0.261 | 0.197 | 0.000 | 0.541 | 0.829 |

| Model | Perturbation | Open Entity | FIGER | TACRED | CosmosQA | GLUE |
|-------|-------------|-------------|-------|--------|----------|------|
| K-Adapter | No Substitution | $76.05 \pm 0.36$ | $77.67 \pm 0.67$ | $70.81 \pm 0.49$ | $80.04 \pm 0.73$ | $87.31 \pm 0.75$ |
| | Ontological Substitution | $76.12 \pm 0.26$ | $77.66 \pm 0.16$ | $70.44 \pm 0.35$ | $79.99 \pm 1.40$ | $88.14 \pm 0.16$ |
| | P-value | 0.754 | 0.978 | 0.254 | 0.951 | 0.060 |

Table 5: Model performance and P-values of t-test on each downstream tasks before and after ontological substitution.

tical significance (0.05). This outcome suggests an absence of substantial evidence to reject the null hypothesis. Therefore, we could not substantiate the existence of a significant difference in model performance before and after factual substitution.

Overall, for language models acquiring factual knowledge from either plain texts or structured knowledge bases, the correctness of factual knowledge during pre-training has a very limited impact on downstream tasks, which significantly challenges the previous assumptions of factual knowledge leading to performance improvements on downstream tasks.

### 4.3 Does Ontological Substitution Affect Downstream Performance?

**Conclusion 3.** *Overall, the influence of ontological substitution is slightly stronger than factual substitution, but still shows very limited impact on most downstream tasks.*

The performance comparison and t-test results about ontological substitution are demonstrated in Table 5. Surprisingly, we find that even type-level ontological substitution still has a limited impact on most downstream tasks, even for tasks that significantly rely on entity type information, such as named entity recognition, relation extraction, and entity typing. The results of the t-test further support this finding. In most downstream tasks for all three models, the p-values exceed the threshold 0.05, indicating a lack of sufficient evidence to confirm a significant difference in model performance before and after ontological substitution. The only exception also comes from FewRel in ERNIE, mainly due to information leakage as mentioned above. Figure 3 illustrates the performance distribution before and after perturbation on several tasks, which illustrates the limited effect of both factual and ontological substitution in a more

straightforward manner. Specifically, we first assume the variation in performance due to random seed effects follows a normal distribution, and plot the performance distribution curve based on the results of the vanilla LMs on various random seeds, then highlight the performance of the model before and after two substitutions on the curve. And we can clearly find that there not exist significant performance difference before and after perturbation. We also notice that, overall, the impact of ontological substitutions is slightly stronger than factual substitutions, which is also understandable because large-scale ontological substitutions not only affect the correctness of factual knowledge in the model but also interfere with the model's understanding of linguistics and semantics, thus undermines the language understanding ability of PLMs.

## 5 Conclusions and Discussions

This paper proposes a counterfactual-based knowledge analysis framework to investigate the influence of injecting factual knowledge into LMs. Throughout our experiments, we find that even though current knowledge injection approaches do inject factual knowledge into models, there exist very limited causal relations between the correctness of factual knowledge and performance improvements. Our findings strongly challenge previous assumptions that the injected factual knowledge is the core reason of previous factual injection approaches to achieve improvements on downstream tasks, but very likely due to other confounding factors.

Our conclusions can also shed light on future research directions. For example, current extremely large LMs such as GPT-3 (Brown et al., 2020) and LLaMA (Touvron et al., 2023) can perform well on various downstream tasks but would still generate large amounts of factual-incorrect responses, and

our conclusions indicate that we need to improve the evaluation paradigm for comprehensive PLM evaluation on factual knowledge.

## Limitations

This paper focuses on factual knowledge injection, in the future, we can further investigate the impact of knowledge injections on other knowledge types such as linguistic knowledge (Ke et al., 2020; Lauscher et al., 2020; Levine et al., 2020; Zhou et al., 2019), syntax knowledge (Zhou et al., 2019; Sachan et al., 2021; Bai et al., 2021) and commonsense knowledge (Bosselut et al., 2019; Guan et al., 2020; Shwartz et al., 2020).

Due to the huge cost and the limitations of computing resources, we have not yet conducted experiments on extremely large language models such as GPT-3 (Brown et al., 2020).

## Acknowledgement

We sincerely thank the reviewers for their insightful comments and valuable suggestions. This work is supported by the Natural Science Foundation of China (No.62122077 and 62106251). Hongyu Lin is sponsored by CCF-Baidu OpenFund.

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

## A  Fine-tuning Details

In the fine-tuning stage, most of the hyper-parameters and model architectures are kept consistent with the original papers. Specifically, for ERNIE and K-Adapter, we strictly follow the original experimental settings. As for BERT, we will introduce our fine-tuning procedure in detail. For *natural language understanding* and *named entity recognition*, we follow Devlin et al. (2019)'s task-specific architecture, which simply incorporates PLMs with an additional output layer. For *open domain question answering* and *fact checking*, we construct the datasets from KILT (Petroni et al., 2021) benchmark. Since we focus on factual knowledge learned by PLMs, we ignore the retrieval stage and instead provide the model with the gold document. For *relation extraction*, we consider PURE (Zhong and Chen, 2021) as our base architecture, which is an approach for utilizing PLMs in relation extraction tasks. In order to avoid error propagation, we evaluate the models with the gold entities. In addition, we use the same data split with Zhong and Chen (2021). The detailed metrics and hyper-parameters are shown in Table 7.

## B  Information Leakage Analysis

As we mentioned in Section 4.2, the performance of ERNIE on FewRel demonstrates the relatively larger performance gap before and after perturbation. To dive into the underlying reasons, we first analyze the knowledge source of ERNIE and FewRel. And we surprisingly find that they share exactly the same knowledge source. Specifically, FewRel is a relation extraction dataset that annotates relations between entities according to Wikidata taxonomy, and the knowledge embedding used by ERNIE is also trained on Wikidata with TransE algorithm. In that case, the relation information of each input entity pair in FewRel is already learned by knowledge embedding passed through ERNIE. This could lead to severe answer leakage and make the model's outputs more rely on the information leaked from the external knowledge source, further leading to the performance gap.

To further verify and quantify the impact of such information leakage, we design a simple non-parametric classification model. Specifically, for each input entity pair in FewRel, we acquire the same corresponding entity embedding with ERNIE, and simply use a k-nearest neighbors algorithm (KNN) for the classification regardless of the input

|  | CoLA | SST-2 | MRPC | STS-B | QQP | MNLI | QNLI | RTE | Avg. |
|---|---|---|---|---|---|---|---|---|---|
| **BERT** | | | | | | | | | |
| Vanilla | 54.29 | 91.58 | 88.70 | 85.74 | 88.48 | 83.84 | 90.16 | 59.28 | $80.26 \pm 0.33$ |
| Factual | 52.64 | 91.74 | 87.87 | 85.72 | 88.37 | 83.78 | 90.06 | 60.36 | $80.07 \pm 0.53$ |
| Ontological | 54.67 | 91.97 | 88.70 | 85.67 | 88.29 | 83.75 | 89.56 | 58.84 | $80.18 \pm 0.36$ |
| **ERNIE** | | | | | | | | | |
| Vanilla | 57.46 | 91.42 | 86.17 | 88.86 | 89.14 | 84.24 | 90.28 | 66.34 | $81.74 \pm 0.59$ |
| Factual | 55.50 | 91.82 | 87.73 | 88.30 | 89.08 | 84.04 | 90.18 | 65.94 | $81.57 \pm 0.34$ |
| Ontological | 55.56 | 91.92 | 87.02 | 88.44 | 89.20 | 84.04 | 90.20 | 66.98 | $81.67 \pm 0.16$ |
| **K-Adapter** | | | | | | | | | |
| Vanilla | 62.87 | 95.99 | 90.34 | 92.01 | 90.67 | 89.91 | 93.97 | 82.82 | $87.31 \pm 0.75$ |
| Factual | 64.58 | 95.96 | 91.12 | 91.81 | 90.68 | 89.88 | 94.31 | 84.84 | $87.90 \pm 0.56$ |
| Ontological | 65.72 | 96.24 | 90.69 | 91.61 | 90.71 | 89.91 | 94.57 | 85.70 | $88.14 \pm 0.16$ |

Table 6: Results on GLUE dev set.

| Dataset | Task | Metric | Hyperparameter | | |
|---|---|---|---|---|---|
| | | | LR | Batch | Epochs |
| GLUE | NLU | * | 2e-5 | 32 | 3 |
| NQ | QA | exact match | 2e-5 | 32 | 5 |
| FEVER | FC | accuracy | 5e-5 | 32 | 5 |
| CONLL2003 | NER | F1 score | 2e-5 | 8 | 5 |
| OntoNotes 5.0 | NER | F1 score | 5e-5 | 32 | 10 |
| ACE2004 | RE | F1 score | 2e-5 | 32 | 10 |
| ACE2005 | RE | F1 score | 2e-5 | 32 | 10 |

Table 7: The metric and hyperparameters of each dataset. *: We follow the metrics mentioned in GLUE (Wang et al., 2019) for each tasks. LR, Batch and Epochs indicate learning rate, batch size and num of epochs, respectively.

| Model | Vanilla | Factual | Ontological |
|---|---|---|---|
| ERNIE | 88.41 | 86.88 | 85.15 |
| KNN | 69.67 | 42.10 | 17.71 |

Table 8: The performance of KNN and ERNIE on FewRel before and after perturbation. Vanilla indicates no perturbation, factual indicates factual substitution, and ontological indicates ontological substitution.

## C GLUE Performance

Table 6 shows the detailed results on GLUE benchmark. We can find that both factual substitution and ontological substitution show limited influence on most tasks, which is consistent with the prior conclusions.

text. Table 8 compares the performance between KNN and ERNIE before and after perturbation. We can see that: 1) The F1 score of KNN without knowledge perturbation achieves 69.67 on FewRel. This model is a simple non-parametric classification model using the same knowledge embedding with ERNIE as input. Such results indicate that the performance of ERNIE may highly rely on the relation information leaked by Wikidata, instead of the knowledge injected in the model. 2) The F1 score of KNN significantly drops from 69.67 to 42.10 after factual perturbation, and further drops to 17.71 after ontological perturbation, which reaches a much larger performance gap caused by knowledge perturbation than ERNIE. Such results indicate that the information leakage is very likely to be the reason why the performance of ERNIE on FewRel demonstrates a relatively larger performance gap before and after perturbation.