# OpenReview forum: "Does the Correctness of Factual Knowledge Matter for Factual Knowledge-Enhanced Pre-trained Language Models?"
_EMNLP/2023/Conference — EMNLP 2023 Main_

### Official Review · Reviewer_jAz2 · 2023-08-01

**Soundness:** 4

**Excitement:**

4: Strong: This paper deepens the understanding of some phenomenon or lowers the barriers to an existing research direction.

**Paper Topic And Main Contributions:**

This paper explores the effect of factuality on methods for factual knowledge injection. In particular, the authors assess:

1. Do knowledge injection methods actually inject knowledge? (yes)
2. Does the factuality of the injected knowledge affect downstream performance on a variety of tasks? (not really)

Experiments are performed with a variety of models that either learn knowledge through the pre-training process (e.g., BERT MLM), or through adding explicit structured data (e.g., ERNIE).

**Reasons To Accept:**

- This paper is a nice sanity check on knowledge-augmented language models. It's always great to see work that makes us rethink existig methods and evaluation protocols.
- This paper is technically sound, and I think the experimental setup is appropriate for the claims

**Reasons To Reject:**

- I'm not totally sure I agree with this framing of the results (and the framing is somewhat unclear). To me, the core takeaway from these results is that usual downstream evaluations don't really assess knowledge at all---you can be injected with factually incorrect knowledge and still do more-or-less fine.
  - similarly, I disagree with the premises in the paper that tasks like NER and open-domain QA really require factual knowledge. These are things that could _benefit_ from factual knowledge (e.g., a gazetteer for NER), but you can usually do just fine on NER with minimal world knowledge as long as you're reasonably good at reading comprehension (which these factually-perturbed models seem to be, since they're still pre-trained on a bunch of english).
- If there was some way of reframing this paper so that it's more centered on the evaluation, I'd like it a lot more. The current evaluations don't really answer the titular question of "Does Factual Knowledge Matter for Factual Knowledge-Enhanced Pre-trained Language Models?"---it ultimately depends on what the downstream task is, and this paper, to me, shows that existing evaluation downstream tasks aren't really meaningful for evaluating whether the model benefits from injecting factual knowledge.
  - Furthermore, the title / much of the paper focuses on _models_, whereas I think this result is a lot more about evaluation / data.

- Anyway, in general, I like this paper and the experiments, so I'd be happy to see it accepted. However, I think the current framing / narrative of the paper somewhat misses the point. It'd be great if the paper could narrow down its claims and make them a bit more forcefully. I know that I can't necessarily force the authors to make these changes, but I really like this paper, and I think it'd be a shame if it were hamstrung by wishy-washy writing. Ultimately just my opinion, though.

**Reproducibility:**

4: Could mostly reproduce the results, but there may be some variation because of sample variance or minor variations in their interpretation of the protocol or method.

**Reviewer Confidence:**

4: Quite sure. I tried to check the important points carefully. It's unlikely, though conceivable, that I missed something that should affect my ratings.

**Typos Grammar Style And Presentation Improvements:**

- Table 2 could be clearer. Maybe arrows between Cicero -> Rome, etc so it's clear what the relations are.
- 435 4.1 Does Knowledge Injection Works? -> 435 4.1 Does Knowledge Injection Work?
- The results tables are terribly confusing (e.g., Table 4). "Vanilla" seems to mean before knowledge perturbation, and "Factual" is after knowledge perturbation? In that case, it's very confusing that the row labeled as "Factual" actually refers to the setting where the injected knowledge is _not_ factual. I get that the "factual" here refers to "factual" vs "ontological" perturbtions, but even something like replacing vanilla with "No Perturbation" and "factual" with "Factual Perturbation" would be a lot more clearer.

---

> ### Author Rebuttal · Authors · 2023-08-29
>
> Firstly, we would like to extend our gratitude for your in-depth review and constructive criticism.
>
> - Regarding the selection of downstream tasks: we have opted for these specific tasks due to prior research, such as ERNIE and K-Adapter, reporting performance enhancements on them through knowledge injection. Therefore, we would like to investigate whether there exist causal relations or just spurious correlations, between knowledge injection and the observed improvements in downstream performance.
> - Regarding the framing of results: firstly, we totally agree with you about "existing evaluation downstream tasks aren't really meaningful for evaluating whether the model benefits from injecting factual knowledge", as well as the suggestion that our writing should focus more on evaluation. Meanwhile, in our opinion, the experimental outcomes in our paper concerning both factual perturbation and ontological perturbation can be attributed to two aspects of reasons.
>     - From a task perspective, as you mentioned, part of the downstream tasks can be accomplished without extensive world knowledge, which are not suitable for exiting knowledge-enhanced LMs evaluation. However, this explanation might not entirely account for the outcomes observed in tasks like RE, as well as the outcomes stemming from ontological perturbation.
>     - From a model perspective, our experiments suggest that current knowledge-enhanced LMs might not be able to effectively leverage knowledge acquired from pre-training to downstream tasks via fine-tuning.
>     - Furthermore, we will take these valuable suggestion into account to revise both our title and paper.
>
> We sincerely thank you for the suggestions about both paper reframing and presentation improvements, which are very insightful and valuable. We will follow the suggestions and carefully revise the paper to make it more precise and comprehensible.

---

### Official Review · Reviewer_tpiB · 2023-08-03

**Soundness:** 3

**Excitement:**

4: Strong: This paper deepens the understanding of some phenomenon or lowers the barriers to an existing research direction.

**Paper Topic And Main Contributions:**

This paper mainly demonstrates whether the so-called knowledge injection in the existing pre-training model really has a real impact on related downstream tasks. The authors introduce a counterfactual-based analysis framework to explore the causal effects of factual knowledge injection on the performance of language models within pretrain-finetune paradigm and find the correctness of injected knowledge only has a very limited effect on the models’ downstream performance.

**Reasons To Accept:**

Throughout the proposed counterfactual-based knowledge analysis framework, the authors find that eventhough current knowledge injection approaches do inject factual knowledge into models, there exist very limited causal relations between the correctness of factual knowledge and performance improvements. This is an interesting discovery, which strongly challenge previous assumptions.

**Reasons To Reject:**

The authors should add some representative work of current knowledge augmented pre-trained models to increase the convincing power of this paper's arguments.

**Reproducibility:**

4: Could mostly reproduce the results, but there may be some variation because of sample variance or minor variations in their interpretation of the protocol or method.

**Reviewer Confidence:**

1: Not my area, or paper was hard for me to understand. My evaluation is just an educated guess.

---

> ### Author Rebuttal · Authors · 2023-08-29
>
> Thank you for your valuable feedback and for pointing out the areas where our paper could be further enhanced.
>
> Regarding the selections of representative knowledge augmented pre-trained models, we would likely to kindly clarify our principle for selecting target models. As we mentioned in section 3.2, currently there are two primary knowledge sources for knowledge acquisition by LMs, including plain text and structural KBs. Therefore, we select 3 representative PLMs including BERT, ERNIE and K-ADAPTER, covering both 2 knowledge sources and 3 distinct knowledge acquisition strategies, and the consistent findings among them demonstrate the generalizability of our conclusions.
>
> Meanwhile, the scope of knowledge-enhanced language models is continuously evolving. And we agree with your suggestion that further incorporating the latest knowledge-enhanced LMs could certainly help solidify our points and broaden the research's impact.
>
> In the future, we will seek more novel knowledge injection strategies and incorporate them into our revisions.

---

### Official Review · Reviewer_3qik · 2023-08-04

**Soundness:** 2

**Excitement:**

4: Strong: This paper deepens the understanding of some phenomenon or lowers the barriers to an existing research direction.

**Paper Topic And Main Contributions:**

This paper aims to check whether the correctness of the knowledge injected by pre-training affects the models' performance on downstream tasks. The authors mainly conducted two types of pre-training, one injects correct knowledge into the models, and the other injects perturbed knowledge. The evaluation results on some downstream tasks show no significant difference between correct knowledge and wrong perturbed knowledge. Based on this observation, the author concludes that the correctness of injected knowledge only has a very limited effect on the models’ downstream performance.

**Reasons To Accept:**

This paper targets an interesting problem and gives insightful results.
The analyzing procedure is well-designed and convictive.

**Reasons To Reject:**

The title is a little misleading since the real problem this paper proposed is "Does *correct* factual knowledge matter..."
If we want to knowledge "Does factual Knowledge matter...," we need to remove the knowledge from models and measure the differences.

The limited scale of the experiments may not support the general conclusions proposed in this paper. For example, do these conclusions hold for other pre-trained language models, such as RoBERTa, and ALBERT?

**Reproducibility:**

4: Could mostly reproduce the results, but there may be some variation because of sample variance or minor variations in their interpretation of the protocol or method.

**Reviewer Confidence:**

2: Willing to defend my evaluation, but it is fairly likely that I missed some details, didn't understand some central points, or can't be sure about the novelty of the work.

---

> ### Author Rebuttal · Authors · 2023-08-29
>
> Thank you for your constructive feedback and thoughtful suggestions, and we would like to clarify the issues you raised.
> - Regarding the title of our paper: in our initial idea, we think injecting incorrect knowledge into LMs also transforms the original factual knowledge within these models into non-existence, hence, we chose the title "Does factual knowledge..." However, your feedback has prompted us to recognize the potential ambiguity associated with this title.  Therefore, we plan to modify the title in the revision to make it more precise and to avoid potential ambiguity.
> - Regarding the scope of our experiments:
>   - Due the huge computational expenses of our experiments (requiring pre-training each LM from scratch twice), we currently opt to select one representative language model for each of the 3 widely used knowledge injection methods, to ensure the generalizability of our conclusions (please refer to section 3.2 for detail).
>   - Given the **shared Transformer architecture, type of knowledge source and the fundamental pre-training task (masked language modeling) among BERT, RoBERTa, and ALBERTA**, we chose BERT as the representative model for learning knowledge from plain text, and we think that our conclusions should remain consistent across these 3 models.
>
> Taking into account your valuable feedback, we will include additional experiments involving other pre-trained language models in our revisions.

---

### Meta-Review · Area_Chair_Aa9b · 2023-09-20

**Recommendation:** 4

**Metareview:**

This paper introduces a novel counterfactual-based analysis framework to investigate the causal effects of factual knowledge injection on pre-trained language model performance within the pretrain-finetune paradigm. The authors challenge existing assumptions by demonstrating that the correctness of injected knowledge has only a limited impact on downstream performance. The findings provide valuable insights and contribute significantly to the understanding of the role of factual knowledge in language models. The reviewers are generally exciting about the insights of this paper. I would therefore recommend to accept this paper to the main conference.

---

### Decision · Program_Chairs · 2023-10-07

**Decision:**

Accept-Main

**Comment:**

This paper introduces a novel counterfactual-based analysis framework to investigate the causal effects of factual knowledge injection on pre-trained language model performance within the pretrain-finetune paradigm. The authors challenge existing assumptions by demonstrating that the correctness of injected knowledge has only a limited impact on downstream performance. The findings provide valuable insights and contribute significantly to the understanding of the role of factual knowledge in language models. The reviewers are generally exciting about the insights of this paper. I would therefore recommend to accept this paper to the main conference.